# Discovery of an Orally Effective Factor IX-Transferrin Fusion Protein for Hemophilia B

**DOI:** 10.3390/ijms21010021

**Published:** 2019-12-18

**Authors:** Chen Xie, Zhijun Wang, Yang Su, Jeffrey Wang, Wei-Chiang Shen

**Affiliations:** 1Department of Pharmaceutical Sciences, College of Pharmacy, Western University of Health Sciences College of Pharmacy, Pomona, CA 91766, USA; cxie@westernu.edu (C.X.); zwang@ketchum.edu (Z.W.); lisasu2107@gmail.com (Y.S.); 2Department of Pharmaceutical Sciences, College of Pharmacy, Marshall B. Ketchum University, Fullerton, CA 92831, USA; 3Department of Pharmacology and Pharmaceutical Sciences, School of Pharmacy, University of Southern California, Los Angeles, CA 90033, USA; weishen@usc.edu

**Keywords:** fusion protein, factor IX, transferrin, oral delivery, hemophilia B

## Abstract

Hemophilia B is a severe blood clotting disorder caused by the deficiency of factor IX (FIX). FIX is not bioavailable when given orally due to poor stability and permeability in the gastrointestinal tract. The feasibility of fusing FIX with transferrin (Tf) to enhance the oral bioavailability of FIX is explored. Seven recombinant fusion proteins (rFIX-Tf) with different linkers were constructed and expressed in HEK293 cells and characterized by in vitro transcytosis and transferrin receptor (TfR) binding assay in Caco-2 cells and a one-stage clotting assay. The in vivo efficacy study was performed using a tail-bleeding model in hemophilia B mice. Fusion proteins rFIX-Tf/G_2_ and rFIX-Tf/SVSQ were most permeable and showed a specific binding ability to TfR in Caco-2 cells. Both proteins retained FIX activity in clotting generation. The in vivo efficacy study showed that both proteins by intravenous injection significantly reduced blood loss. Most significantly, rFIX-Tf/G_2_ demonstrated anti-bleeding activity when administered orally. Our results showed that the fusion protein technique with Tf could be potentially used for oral delivery of FIX and the linker between FIX and Tf in the fusion protein is crucial. rFIX-Tf/G_2_ appears to be the most promising fusion protein as potential oral therapeutics for hemophilia B.

## 1. Introduction

Coagulation factor IX (FIX, Christmas factor), encoded by the gene located on the X chromosome (Xq27.1-q27.2), is a serine protease essential for hemostasis with X-linked recessive inheritance [1]. It was discovered in 1952 after a young boy named Stephen Christmas was found having hemophilia due to the lack of this factor [2]. Complete deficiency of FIX results in a severe bleeding disorder known as hemophilia B or Christmas disease. Hemophilia B is a lifelong disease, affecting approximately 1 in 25,000 to 30,000 live male births [3]. FIX has 415 amino acids in its sequence and a molecular weight of approximately 57 kDa and is synthesized in the liver and secreted as a zymogen (an inactive pro-enzyme) into the bloodstream [4]. It is converted to the active form (FIXa) with two chains linked by a disulfide bond via minor proteolysis by either factor IXa (contact pathway) or factor VIIa (tissue factor pathway) [5]. In the presence of calcium ions, membrane phospholipids and a factor VIII cofactor, FIXa, activates the coagulation cascade by converting factor X to factor Xa. 

In healthy individuals, the plasma concentration of FIX is around 5 µg/mL [6]. The standard care for hemophilia B patients is the replacement therapy with plasma-derived or recombinant FIX (rFIX). However, the short elimination half-life due to the rapid in vivo clearance of FIX makes multiple injections necessary. FIX products with an extended half-life can potentially reduce the number of injections per bleeding episode and improve patient care. Indeed, efforts in this arena have been made based on PEGylation, and albumin or Fc fusion approaches [7,8,9,10,11,12]. 

Currently, all commercial FIX products for hemophilia patients are given by injection, which is inconvenient. The development of the oral FIX product can greatly improve patient compliance. However, oral protein delivery is challenging because proteins are usually not bioavailable orally due to the poor stability and permeability in the gastrointestinal tract [13]. 

Receptor-mediated transcytosis has been proven to a potential approach to deliver large bioactive molecules administrated orally, which otherwise have negligible bioavailability [14,15]. Serum transferrin (Tf) is a glycoprotein of approximately 76.5 kDa in size whose primary function is to bind and transport non-heme iron via Tf receptor (TfR) [16]. Previous studies have demonstrated that TfR could be used to increase epithelial absorption of protein and peptide drugs [15]. Our group has shown that Tf and its chemical conjugates or fusion proteins could be transported across various cultured epithelial cell monolayers and enhance oral bioavailability [17,18,19]. For example, the insulin-Tf conjugate showed a hypoglycemic effect in diabetic rats [20,21] and the G-CSF-Tf conjugate exhibited myelopoietic activity in BDF-1 mice following oral drug administration [22]. More recently, a recombinant G-CSF-Tf fusion protein was successfully produced with both TfR binding ability in Caco-2 cells and the proliferative activity in NFS-60 cells [23]. The G–CSF–Tf fusion protein was also found to be absorbable across gastrointestinal epithelia, as indicated by its myelopoietic activity when orally administered in BDF-1 mice [24]. Similar results were observed with human proinsulin and growth hormone-Tf profusion proteins [25,26,27].

These results suggested that the fusion protein, by fusing the therapeutic protein with Tf, could be a promising technology for oral protein or peptide delivery. Therefore, in this study, several rFIX and Tf fusion proteins are developed using the recombinant DNA approach and their in vitro and in vivo activities are investigated. 

## 2. Results 

### 2.1. Construction, Preparation and Quantification of FIX Fusion Proteins

#### 2.1.1. FIX-Tf Expression Constructs and Fusion Protein Expression

Seven fusion proteins with various space peptides were constructed and expressed in HEK293 cells. The final expression constructs were confirmed by enzymatic digestion (Appendix A) and DNA sequencing. Under non-reducing conditions, SDS-PAGE analysis demonstrated that FIX and Tf expressed as a single, dominant band with about 90% abundance and about 130 kDa molecular weight corresponding to the fusion protein, which is the sum of FIX (57 kDa) and Tf (79 kDa) (Figure 1A). No protein at about 130 kDa could be detected in concentrated supernatants from cells transfected with pCDNA3.1(+) (negative control). According to the FIX:Ag level by ELISA, the expression level of fusion proteins with non-cleavable linkers was much higher than that of fusion proteins with cleavable linkers. FIX-Tf/G2 was found with the highest level (Table 1).

To confirm the identity of the fusion proteins, the conditioned medium for each fusion protein was analyzed using Western blot and probed with both anti-FIX and anti-Tf antibodies. The data revealed that both antibodies detected the fusion proteins with about 130 kDa molecular weight (Figure 1B).

#### 2.1.2. Purification of Recombinant Fusion Protein by Size Exclusion Chromatography

The chromatograms of the SEC are shown in Appendix A. Samples before and after purification were applied to SDS-PAGE and proteins were visualized by Coommassie blue staining (Appendix A). The purified protein gave one band at a molecular mass of 130 kDa under reducing condition. About 90% of the fusion protein was lost after purification due to the many steps involved in the processing. Since the purity of the fusion protein expressed in the large culture dishes (150 cm^2^) was high enough even before purification, we used the fusion proteins without purification for subsequent in vitro and in vivo studies.

### 2.2. In vitro Activity Studies

#### 2.2.1. Apical-to-Basolateral Transcytosis of BeneFIX^®^ and FIX–Tf Fusion Proteins Across Caco-2 Cell Monolayers

BeneFIX^®^, a marketed coagulation factor IX (recombinant) product, was utilized as the control. Two-week-old Caco-2 monolayers, with TEER of approximately 500 Ω∙cm^2^, were dosed with about 100 mIU/mL BeneFIX^®^ and various FIX-Tf fusion proteins in the apical side of 6-well transwells. As shown in Figure 2, the FIX-Tf fusion proteins with a non-cleavable linker named FIX-(GGGGS)_2_-Tf (FIX-Tf/G_2_) exhibited the highest transport rate across Caco-2 cell monolayers at 2, 4, and 6 h, respectively. The amount of transported FIX-Tf/G_2_ was 7.8-fold higher than BeneFIX^®^ after a 6-h incubation (0.187% of protein transport rate for FIX-Tf/G_2_ and 0.024% of protein transport rate for BeneFIX^®^). In addition, the transport rates for two FIX fusion proteins with cleavable linker were also significantly higher than the BeneFIX^®^ at 6 h (*p* < 0.05), with FIX- SVSQTSKLTRAETVFPDVDGS-Tf (FIX-Tf/SVSQ) transported 5.8-fold higher than BeneFIX^®^ and FIX-dithiocyclopeptide-Tf (FIX-Tf/Dithi) transported 4.4-fold higher than BeneFIX^®^. When compared to FIX-Tf, FIX-Tf/SVSQ and FIX-Tf/Dithi transported 1.6 and 1.5 folds higher, respectively. The TEER of Caco-2 cells was not significantly changed during the 6-h treatment, indicating good integrity of the monolayers.

#### 2.2.2. In vitro Tf Receptor binding activity of the FIX–Tf fusion proteins

To evaluate the Tf activity of the fusion proteins, the binding capability of the fusion proteins to TfR in Caco-2 cells was assessed. The TfR binding competition assay was performed in the presence of 3 µg/mL Tf with different doses of fusion proteins. As shown in Figure 3, the higher the amount of fusion proteins, the greater the binding of Tf to TfR in cultured Caco-2 cells. The positive correlation indicated that the fusion protein maintained a specific binding ability to TfR. It also showed that BeneFIX^®^ has no TfR binding activity, as expected. 

#### 2.2.3. Activated Thromboplastin (aPTT) Time Assay for in Vitro Clotting Activities of Fusion Proteins

A one-stage clotting assay with a PTT reagent is most widely used for in vitro FIX clotting activity. Their activities were related to the optical density of the solutions which were determined as the difference at 280 and 320 nm as a measure of protein content (Table 2). Among the different rFIX-Tf preparations, the clotting activity of rFIX-Tf with non-cleavable (GGGGS)_2_ was higher than others (rFIX-Tf/G_2_ > rFIX-Tf/G_5_ > rFIX-Tf/SVSQ > rFIX-Tf/A_2_ > rFIX-Tf/A_5_ > rFIX-Tf > rFIX-Tf/Dithi). 

### 2.3. In Vivo Studies

#### In Vivo Efficacy in Hemophilia B Mice

The efficacy of different fusion proteins in treating acute bleeding was evaluated in a tail clip bleeding model by intravenous injection. As shown in Figure 4A, there was much less blood loss in wild-type mice than in hemophilia B mice (*p* < 0.001). The positive control BeneFIX^®^ showed good activity in reducing blood loss. The treatment of 50 IU/kg and 20 IU/kg rFIX-Tf/G_2_ significantly reduced blood loss in comparison to the vehicle control in a dose-dependent manner. Thereafter, we used the dose of 20 IU/kg for efficacy comparison. rFIX-Tf/G_2_ and rFIX-Tf/SVSQ at 20 IU/kg both significantly reduced blood loss without significant difference. The results indicated that these two fusion proteins were effective in treating acute bleeding in hemophilia B mice following intravenous injection. 

For the oral efficacy study, it showed that oral 200 IU/kg rFIX-Tf/G_2_ treatments in hemophilia B mice significantly reduced blood loss in comparison to the vehicle control. In comparison, oral rFIX-Tf/SVSQ and BeneFIX^®^ at 200IU/kg had no significant effect on treating acute bleeding with oral administration (Figure 4B). 

## 3. Discussion

In this article, FIX was fused with Tf, and the feasibility of using transferrin as a carrier for oral delivery of rFIX was investigated. To our knowledge, this is the first study using Tf-based recombinant technology to deliver rFIX orally. Seven recombinant fusion proteins consisting of both human FIX and human Tf moieties were engineered and expressed in HEK293 cells. Two fusion proteins were selected due to their promising effect in vitro and favorable yielding rate. One fusion protein showed potential oral activity when compared to the control (BeneFIX^®^). 

The purification of protein drugs is another challenge. The weakness of this study is that we used the fusion proteins directly concentrated from culture medium for subsequent in vitro and in vivo studies without further purification. We tried to use size exclusion chromatography to purify fusion protein due to the good separation with a minimum volume of elute and preservation of activity of protein drugs. However, 90% of the fusion protein was found lost after purification due to excessive processing steps such as buffer exchanging and concentrating. Because the purity of the fusion protein expressed in the large culture dishes (150 cm^2^) was high enough before purification and this project is a proof-of-concept study, the fusion proteins directly concentrated from culture medium were used for the subsequent studies.

One important factor for the successful construction of the fusion protein is the selection of a proper linker to connect different proteins [28]. Fusion proteins generally consist of stable peptide sequences, including the glycine-serine linker (GGGGS)_n_ and α-helix–forming peptide linkers (A(EAAAK)_n_A) (*n* = 2–5), which can provide structure flexibility, improve protein stability, or increase biological activity [29,30,31]. The linker can be cleavable and non-cleavable. Cleavable linkers can be digested in vivo leading to the release of the original functional proteins, while non-cleavable linkers are stable in vivo. In this study, four non-cleavable linkers including two glycine-serine linkers ((GGGGS)_2_ and (GGGGS)_5_) and two α-helix–forming peptide linkers (A(EAAAK)_2_A and A(EAAAK)_5_A) were designed. Non-cleavable linkers do not allow the separation of the two fusion protein domains in vivo. They have several limitations including steric hindrance between two functional domains, altered bio-distribution and metabolism of the protein moieties due to interference with each other, and incorrect folding of the fusion protein. Therefore, additional two cleavable linkers that allow the removal of Tf in vivo were incorporated to increase the specific activity of FIX. One linker is the dithiocyclopeptide linker [32], while the other one is a linker derived from the natural activation peptide of human FIX with the sequence SVSQTSKLTRAETVFPDVDGS [9]. In addition, another fusion protein with a dipeptide Leu-Glu (LE) between the FIX and Tf as a short connection was created by introducing the restriction recognition site of Xhol for the construction of expression vector. 

The linkers between Tf and FIX were found essential to the protein expression level, receptor binding, transcytosis as well as in vitro and in vivo activities. The expression level of fusion proteins with non-cleavable linkers was much higher than that of cleavable linkers, with the highest yielding rate found in FIX-Tf/G_2_. The apical-to-basolateral transcytosis study results showed that rFIX-Tf fusion proteins with non-cleavable linker (GGGGS)_2_ (rFIX-Tf/G_2_) have the highest transport rate across Caco-2 cell monolayers (7.8-fold higher than rFIX), The fusion protein with cleavable linker SVSQTSKLTRAETVFPDVDGS and dithiocyclopeptide also transported 5.8-fold and 4.4-fold higher than rFIX (BeneFIX^®^), respectively. These results suggested that these fusion proteins may be absorbed in the gastrointestinal tract. However, the rate of TfR-mediated transcytosis in intestinal epithelial cells was low, which is most likely due to the predominant basolateral distribution of TfR in intestinal epithelial cells [33]. 

Furthermore, the TfR bind capacity for fusion protein with different linkers was investigated. TfR binding of fusion proteins is a crucial step in oral delivery that enables gastrointestinal absorption via Tf-TfR mediated transcytosis. It was shown that different dosages of fusion proteins could recognize and competitively bind to TfR in the presence of 3 µg/mL Tf on Caco-2 cells. The capability of affinity binding to TfR by the fusion proteins indicated that Tf retained the binding activity to TfR in form of fusion proteins. The in vitro clotting assay further ranked the relative activity of the fusion proteins, and FIX-Tf/G_2_ exhibited the highest potency. The activities between the fusion proteins with commercial rFIX product BeneFIX^®^ which was dissolved in 0.9% NaCl cannot be compared directly because of the existence of interference from the culture medium. When we dissolved BeneFIX^®^ in the same culture medium, it also gave a very low activity (data not shown). The measured activities of fusion proteins could be much lower than those of their real activities. Therefore, only the relative activity of the fusion proteins was compared among the fusion proteins.

Based on the promising in vitro results, rFIX-Tf/G2 (with non-cleavable linker) and rFIX-Tf/SVSQ (with cleavable linker) were selected for subsequent in vivo efficacy using FIX(−/−) mice. Results for in vivo experiments of treating acute bleeding in hemophilia B mice confirmed the in vitro studies. It showed that both rFIX-Tf/G_2_ and rFIX-Tf/SVSQ fusion proteins at a dosage of 20 IU/kg displayed a good in vivo bioactivity by intravenous injection. A higher dosage of 50 IU/kg rFIX-Tf/G_2_ treatment exhibited a better activity in a dose-dependent manner to reduce blood loss to the same level in normal wild type mice.

The in vivo efficacy study was also performed for oral administration and found out that oral rFIX-Tf/G_2_ elicited a significant effect on reducing blood loss in mutant hemophilia B mice, whereas oral rFIX-Tf/SVSQ was ineffective. Although the rate of TfR-mediated transcytosis in intestinal epithelia cells was low, oral rFIX-Tf/G_2_ was effective in reducing blood loss. This result indicated that the insertion of a glycine-serine linker (GGGGS)_2_ in the fusion protein may lead to an effective drug delivery system for FIX by enhancing bioactivity of the rFIX-Tf fusion protein. In this study, a dose ten times (200 IU/kg) higher than IV dose was used and the efficacy was tested 18–20 min post-admission. To verify the oral activity of rFIX-Tf/G_2_, the efficacy study could be further investigated to characterize the dose–response relationship and the time-course of the rFIX-Tf/G_2_ fusion protein, following both intravenously and oral administrations. Nonetheless, oral administration of the fusion protein rFIX-Tf/G_2_ elicited a good in vivo activity, which indicated the feasibility of oral delivery of FIX for hemophilia B patients. 

Fusion protein has been used in the biopharmaceutical industry for several years to improve the pharmacokinetics properties especially half-life extension [34]. Two long-acting recombinant FIX-fusion proteins named ALPROLIX^®^ (fused with Fc) and IDELVION^®^ (fused with albumin) have successfully been developed with prolonged elimination half-life compared to that of conventional plasma-derived FIX (pdFIX) or recombinant FIX (rFIX) product. These products can significantly reduce the number of bleeding episodes [35,36] by intravenous administration. Therefore, it is worth further investigating the pharmacokinetic aspects of fusion proteins rFIX-Tf/G_2_ by intravenous and oral administration in mutant hemophilia B mice in the future.

## 4. Materials and Methods

### 4.1. Construction, Preparation and Quantification of FIX Fusion Proteins

#### 4.1.1. Cloning of Recombinant FIX Fusion Proteins

FIX wild-type cDNA was prepared for genetic fusion to transferrin by introducing a restriction site Xhol replacing the natural FIX stop codon. Briefly, the human FIX was synthesized by Gencript USA Inc. (Piscataway, NJ, USA), which was designed to cover the whole completing coding sequence but without the stop codon (TAA). The restriction recognition sites of AFLII and Xhol were added. The coding area of human FIX was amplified by PCR using primers 5’-CTTAAGACCACTTTCACAATCTGCTAG-3’ and 5’- CTCGAGAGTGAGCTTTGTTTTTTCCT-3’. The FIX-specific primers were incorporated with Xhol and AFLII restriction enzyme sites. The coding area of transferrin without signal peptide sequence was amplified by PCR from the plasmid TFR27A (American Type Culture Collection, Manassas, VA, USA) using primers 5’-ACCGCTCGAGGTCCCTGATAAAACTGTGAGAT-3’ and 5’- GTAGTCTAGATTAAGGTCTACGGAAAGTGCA-3’. The Tf-specific primers were incorporated Xhol and XbaI restriction enzyme sites. The transferrin fragment was cloned into pCDNA3.1(+). Then the Tf fragment was digested with restriction endonucleases Xhol and XbaI and ligated into Xhol/XbaI digested FIX-pCDNA3.1(+). A dipeptide Leu-Glu was created between the FIX and Tf as a short connection due to the cloning site Xhol. The sequence of FIX-Tf-pCDNA3.1(+) was confirmed by DNA sequencing. The plasmid encoding FIX-Tf with different linkers were constructed using a similar method as described above. The information about the linkers is listed in Table 3. 

#### 4.1.2. Expression of Fusion Proteins 

HEK293 cells were seeded into 150 cm^2^ cell culture dishes (BD Biosciences, Franklin Lakes, NJ, USA) the day before transfection. When 80–90% confluence was achieved, the medium was replaced with serum-free and antibiotics free EMEM medium. The plasmids encoding the fusion proteins were transfected into HEK293 cells using linear polyethylenimine (Polysciences, Warrington, PA, USA). After a 5–6-h incubation, the transfection mixture was removed and replaced with conditioned serum-free CD293 medium supplemented with 1 % pen/strep, 10 μg/mL Vitamin K1, and 4 mM L-Glutamine. The conditioned media containing the fusion protein was collected twice every 3 days and centrifuged at 4000× *g* for 25 min at 4 °C. The supernatant was further concentrated by tangential flow filtration with a molecular mass cut off of 30kD (TFF; Millipore, Billerica, MA, USA) to a final volume of 20 mL. After a final buffer exchange to a buffer containing 10 mM histidine, 260 mM glycine, 1% sucrose, 0.005% Tween 80, the concentrated fusion protein was stored at −80 °C until use.

#### 4.1.3. Characterization of the Fusion Proteins by SDS-PAGE and Western Blot Analysis

Samples were denaturated by incubating at 95 °C for 5 min in a non-reduced sample buffer and separated by using SDS-PAGE (pre-cast 8% × 16% gradient gels, Biorad, Hercules, CA, USA). The gels were then stained with 0.1% Commassie blue. To determine the presence of FIX and Tf domain in the fusion protein, anti-FIX and anti-Tf Western blot was performed. Antibody against human FIX (ab124815, Abcam, Cambridge, MA, USA) and antibody against human Tf (HPA005692, Sigma-Aldrich, St. Louis, MO, USA) were used as the primary antibodies, while horseradish peroxidase-conjugated anti-rabbit IgG antibody (#7074, Cell Signaling, Danvers, MA, USA) was used as the secondary antibody. The peroxidase activity was detected by maximun sensitivity chemiluminescence (#34095, Thermo Scientific, Waltham, MA, USA) for visualization.

#### 4.1.4. Purification of Recombinant FIX Fusion Protein by Size Exclusion HPLC

The fusion proteins were purified by size exclusion chromatography (SEC) using an AKTA Purifier UPC 100 system (GE Healthcare, Wauwatosa, WI, USA) with a Hiprep 26/60 Sephacryl S-200 HR column (60 cm × 26 mm) (GE Healthcare, Wauwatosa, WI, USA). The proteins were eluted with 30 mM Tris, 100 mM NaCl, pH 6.8. The flow rate was 1.0 mL/min and the detection wavelength was set at 280 nm. The protein eluted in the first peak was collected and concentrated on Amicon Ultra-15 centrifuge filter units with a molecular mass cutoff of 50 kDa (Millipore, Billerica, MA, USA). After a final buffer exchange to a buffer containing 10 mM histidine, 260 mM glycine, 1% sucrose, 0.005% Tween 80, the purified fusion protein was stored at −80 °C until use.

#### 4.1.5. Human FIX ELISA Assay

The quantification of these fusion proteins was carried out using a FIX ELISA kit (AssayPro, Charles, MO, USA) according to the manufacturer’s instructions. Briefly, the antibody specific for FIX was pre-coated on a 96-well microplate followed by a 2-h incubation with samples and a serial dilution of FIX standards. The samples were sandwiched by the immobilized antibody and biotinylated polyclonal antibody specific for FIX, which was recognized by horseradish peroxidase conjugate. Finally, a peroxidase enzyme substrate was added for detection and absorbance was read at 450 nm. Concentrations of test samples were calculated using the human FIX standard as a reference. This assay was used to measure human FIX antigen (FIX:Ag) in cell culture supernatant and plasma and did not have cross-reactivity with mouse FIX. 

### 4.2. In Vitro Activity Studies

#### 4.2.1. TfR-Mediated Apical-to-Basolateral Transcytosis of Fusion Proteins Across the Caco-2 Cell Monolayer

Caco-2 cells were grown on 0.4 μm pore size polycarbonate filters in Transwells (Costar, Cambridge, MA, USA) for 2 weeks to develop the tight junction. The cell monolayer was then washed once with DMEM containing 0.1% BSA and incubated at 37 °C for 45 min to deplete endogenous Tf. BeneFIX^®^ (FIX control) and various fusion protein preparations were loaded in the apical compartment (100 mIU/mL). Nonspecific transport was also measured in parallel by the inclusion of 100-fold excessive Tf. At 2, 4, and 6 h post-dosing, 500 μL samples were collected from the basolateral compartment and replenished with an equal volume of fresh DMEM. The extent of TfR-mediated transcytosis was determined by subtracting nonspecific transport from total transport. The transported proteins in the basolateral media were determined on the basis of FIX antigen measured with the FIX ELISA assay. The integrity of the cell monolayer was monitored during the experiment by measuring transepithelial electrical resistance (TEER). 

#### 4.2.2. TfR Binding Affinity of Fusion Proteins

Caco-2 cells were seeded in 12-well cluster plates and cultured for 2 weeks until full differentiation. Caco-2 monolayer was washed with cold PBS three times and incubated in serum-free DMEM supplemented with 0.1% bovine serum albumin (BSA) at 37 °C for 30 min to remove any endogenous Tf. A mixture of 3 μg/mL Tf with 3, 10 or 30 μg/mL of BeneFIX^®^ or various FIX-Tf fusion proteins in serum-free DMEM supplemented with 0.1% BSA were added to different wells. After 30 min of incubation at 4 °C, the medium was removed, and the cells were washed with cold PBS three times. The cells were then be lysed using 1 M NaOH and transferrin in the cell lysates were counted by a human FIX ELISA kit from AssayPro (Charles, MO, USA). The total protein in the cell lysate was measured and used to normalize the data to per mg cell protein.

#### 4.2.3. Activated Thromboplastin (aPTT) Time Assay

In vitro clotting activities of FIX in fusion proteins were determined using a commercially available activated thromboplastin time (aPTT) assay. FIX-deficient human plasma was obtained from Aniara Diagnostica (West Chester, OH, USA), and aPTT reagent Pathromtin SL got from Siemens Healthcare Diagnostics (Los Angeles, CA, USA). For sample measurement, the FIX-deficient plasma was supplemented with BeneFIX^®^ or various FIX-Tf fusion proteins to final concentrations of 6.25%–100% FIX. Samples were evaluated against a standard curve prepared with FIX standards.

### 4.3. In Vivo Studies

#### 4.3.1. Efficacy Study Using a Tail-Bleeding Mouse Model of Hemophilia B

Among the seven rFIX-Tf fusion proteins, two most promising fusion proteins, rFIX-Tf/G_2_ (with non-cleavable linker) and rFIX-Tf/SVSQ (with cleavable linker), were selected for in vivo efficacy and pharmacokinetic studies according to (1) the protein expression level, (2) the transport rate across Caco-2 cells, (3) transferrin receptor binding activity, and (4) in vitro clotting activity. BeneFIX^®^ was used in these studies as a control. All doses are expressed as a rFIX-equivalent unit and selected based on relevant published studies.

For the assessment of in vivo efficacy of rFIX-Tf fusion proteins, the mutant hemophilia B male mice (B6.129P2-F9tm1Dws/J strain) were used [37,38] with the wild type male mice as the control. All the animals were housed in a controlled temperature of 21–23 °C room with a 12 h light-dark cycle where food and water were available ad libitum. The animal protocol was approved by the Institutional Animal Care and Utilization Committee (IACUC) at Western University of Health Sciences. The mice were anesthetized with isoflurane and placed on a heating pad to maintain body temperature. The FIX-Tf fusion proteins, BeneFIX^®^ or vehicle solution were injected via tail vein at doses of 50 IU/kg or 20 IU/kg for fusion protein and BeneFIX^®^, respectively. After 5 min, the distal 4 mm of the tail was clipped. Blood was collected into 13 mL of saline (37 °C) for 15 min. Blood loss will be determined by quantifying the amount of hemoglobin in the 15-min collection sample. Briefly, the red blood cells were separated by centrifugation and lysed with hemoglobin reagent (Sigma-Aldrich, St. Louis, MO, USA). The absorbance at 540 nm was measured for the measurement of hemoglobin. The total amount of hemoglobin was determined from a standard curve. For oral efficacy study, a dose 10 times of selected IV dose (200 IU/kg) was used for oral gavage. The duration of the efficacy test was 18–20 min post-administration.

#### 4.3.2. Statistical Analysis

Data were analyzed using GraphPad PRISM version 6 (La Jolla, CA, USA) and presented as mean ± SEM. The difference among various groups was compared using 1-way ANOVA, and *p* < 0.05 was considered statistically significant.

## 5. Conclusions

Seven rFIX-Tf fusion protein expression constructs with different linkers between the FIX and Tf moieties were engineered, successfully produced in HEK293 cell, and characterized by an array of in vitro and in vivo experiments. The fusion proteins showed the ability to be transported through the Caco-2 monolayers, an in vitro model for gastrointestinal drug absorption. More importantly, this fusion protein demonstrates anti-bleeding activity in a mouse model of hemophilia B when administered both intravenously and orally. Thus, this project provides the evidence for the feasibility of using transferrin (Tf) based fusion protein as an approach to develop an oral FIX dosage form for hemophilia B patients. The fusion protein with a stable helix linker G_2_ (rFIX-Tf/G_2_) appears to be the most promising fusion protein as a potential oral therapeutics for hemophilia B.

## Figures and Tables

**Figure 1 ijms-21-00021-f001:**
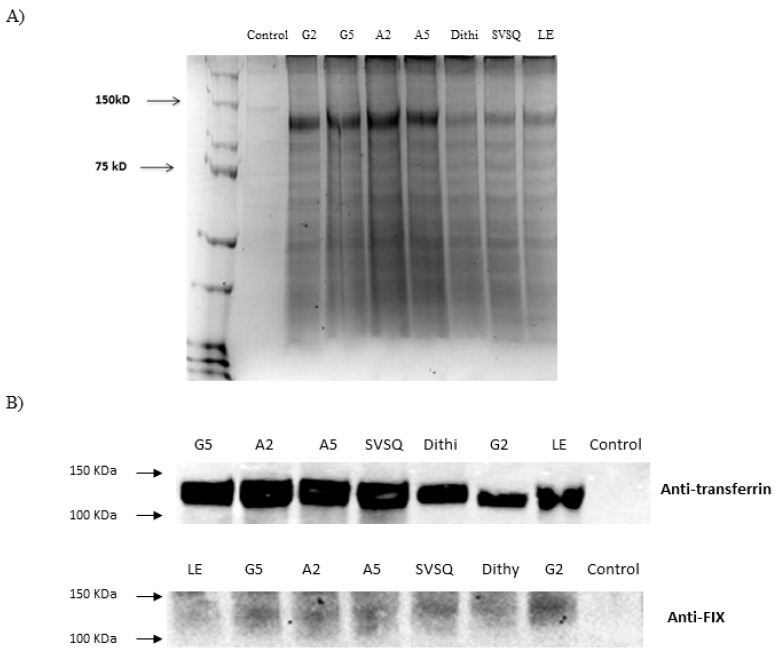
Identification of different fusion proteins: (**A**) SDS-PAGE of recombinant fusion proteins expressed in culture dishes. G2: rFIX-Tf/G_2_; G5: rFIX-Tf/G_5_; A2: rFIX-Tf/A_2_; A5: rFIX-Tf/A_5_; Dithi: rFIX-Tf/Dithi; SVSQ: rFIX-Tf/SVSQ; LE: rFIX-Tf. (**B**) Western blot of recombinant fusion proteins using anti-transferrin and anti-FIX antibodies. Both antibodies detected the fusion protein with about 130 kDa molecular weight.

**Figure 2 ijms-21-00021-f002:**
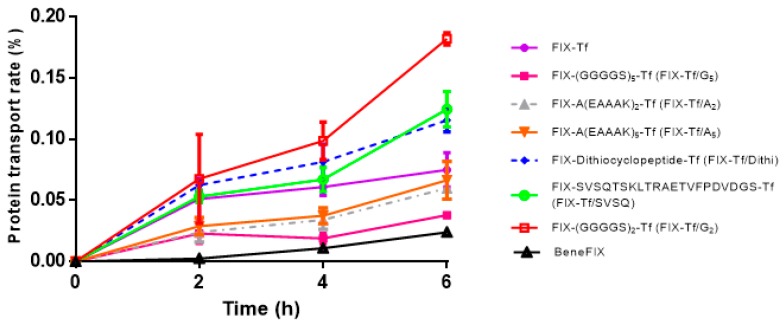
Specific apical-to-basolateral transcytosis of BeneFIX^®^ and FIX–Tf fusion proteins across Caco-2 cell monolayers. Each point represents the mean ± SEM (*n* = 3).

**Figure 3 ijms-21-00021-f003:**
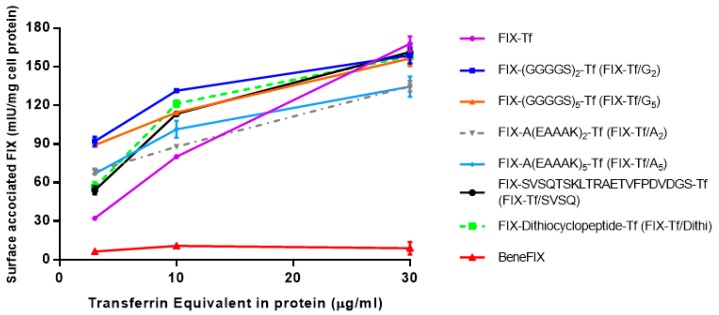
TfR-binding affinity of various Tf-FIX fusion proteins and BeneFIX^®^ in Caco-2 cells. Each point represents the mean ± SEM (*n* = 3).

**Figure 4 ijms-21-00021-f004:**
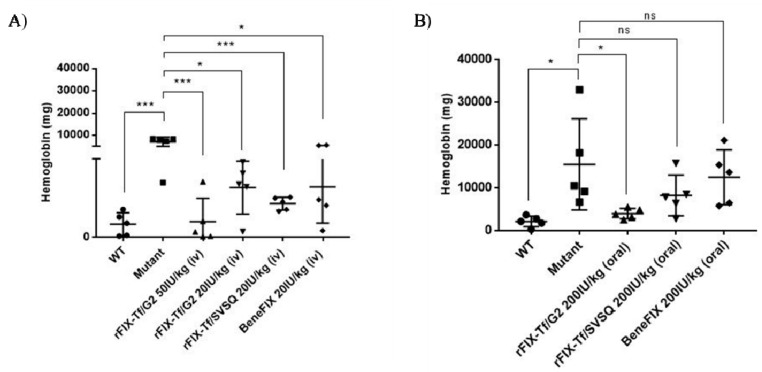
The efficacy of fusion proteins and BeneFIX^®^ in treating acute bleeds in hemophilia B mice. (**A**) The in vivo efficacy by intravenous injection. (**B**) The in vivo efficacy by oral administration. The blood loss was determined by quantifying the amount of hemoglobin for each group. * *p* < 0.05, *** *p* < 0.001, ns = not significant (*p* > 0.05).

**Table 1 ijms-21-00021-t001:** Expression level of FIX:Ag in 150 cm^2^ dish.

Sample Designation	Expression of FIX:Ag in One 150 cm^2^ Dish (IU/Dish)
rFIX-Tf/G_2_	5.81
rFIX-Tf/G_5_	4.04
rFIX-Tf/A_2_	4.50
rFIX-Tf/A_5_	4.78
rFIX-Tf/Dithi	1.04
rFIX-Tf/SVSQ	1.82
rFIX-Tf	1.76

**Table 2 ijms-21-00021-t002:** In vitro clotting activity of fusion proteins.

Sample Designation	FIX Clotting Activity/OD280–320 (IU/OD)
rFIX-Tf	0.075
rFIX-Tf/G_2_	0.199
rFIX-Tf/G_5_	0.185
rFIX-Tf/A_2_	0.096
rFIX-Tf/A_5_	0.094
rFIX-Tf/Dithi	0.073
rFIX-Tf/SVSQ	0.118

**Table 3 ijms-21-00021-t003:** The construct of fusion proteins.

Sample Designation	Linker Type	Linker Sequence *
rFIX-Tf	Dipeptide	LE
rFIX-Tf/G_2_	Non-cleavable	(GGGGS)_2_ + LE
rFIX-Tf/G_5_	Non-cleavable	(GGGGS)_5_ + LE
rFIX-Tf/A_2_	Non-cleavable	A(EAAAK)_2_A + LE
rFIX-Tf/A_5_	Non-cleavable	A(EAAAK)_5_A + LE
rFIX-Tf/Dithi	Cleavable	Dithiocyclopeptide + LE
rFIX-Tf/SVSQ	Cleavable	SVSQTSKLTRAETVFPDVDGS + LE

*** LE:** A dipeptide Leu-Glu (LE) between the FIX and Tf as a short connection due to the cloning site Xhol.

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
