# Peer review of "Discovery of an Orally Effective Factor IX-Transferrin Fusion Protein for Hemophilia B"

_ijms, 2019, doi:10.3390/ijms21010021_

Round 1
Reviewer 1 Report
The authors show in this manuscript a study where in vitro and in vivo activities of several rFIX and Tf fusion proteins were investigated, which were developed by the authors. The text is clear and shows interesting results.
Consequently, this work should be suitable for publication in International Journal of Molecular Sciences. Minor changes have to take into account:
Please, indicate what is BeneFIX© in the section “Results” instead of the section “Discussion”.
The Y axis scales (haemoglobin mg) in the Figure 4A and Figure 4B are quite different, therefore is difficult to compare the results between oral and iv exposure. Furthermore, the result of mutant seems to be higher in oral exposure than iv exposure and it is difficult to compare the WT results. Please, use the same Y axis scales in both figures.
Please, indicate the dose in mg/kg body weight in addition to of IU/kg body weight in section “Materials and Methods”, subsection “4.3. In vivo studies”.
In page 6, lines 192-193, I suppose the authors meant “… transported 5.8-fold and 4.4-fold higher than rFIX, respectively…” instead of “…transported 5.8-fold and 4.4-fold higher than rFIX-Tf, respectively…”.
Author Response
The authors show in this manuscript a study where in vitro and in vivo activities of several rFIX and Tf fusion proteins were investigated, which were developed by the authors. The text is clear and shows interesting results.
Consequently, this work should be suitable for publication in International Journal of Molecular Sciences. Minor changes have to take into account:
Please, indicate what is BeneFIX© in the section “Results” instead of the section “Discussion”.
Response: Thank you for the comment. We have made the changes accordingly.
The Y axis scales (haemoglobin mg) in the Figure 4A and Figure 4B are quite different, therefore is difficult to compare the results between oral and iv exposure. Furthermore, the result of mutant seems to be higher in oral exposure than iv exposure and it is difficult to compare the WT results. Please, use the same Y axis scales in both figures.
Response: Thank you for the comment. In this study, the hemoglobin levels were higher in the oral groups, so if we used the same scale, the data of iv groups would be too concentrated on the bottom. In order to accommodate your request, we changed the y axis into two segments (see the revised version of Figure 4).
Regarding the discrepancy of hemoglobin levels in the mutant groups between iv and oral doses, it could be due to the large variation in the mutant mice. In the oral study, two mutant mice happened to produce much higher hemoglobin levels (~18000 and 32000 mg), which led to a higher average than i.v. groups. However, these two values were still within 2 SD range and we could not exclude them for data analysis.
Please, indicate the dose in mg/kg body weight in addition to of IU/kg body weight in section “Materials and Methods”, subsection “4.3. In vivo studies”.
Response: Thank you for the comment. In this study, the recombinant fusion proteins were quantified using units (reflecting the activity) instead of the absolute weight. This is a similar approach of dose labeling to the commercial recombinant coagulation factor IX (BeneFIX).
In page 6, lines 192-193, I suppose the authors meant “… transported 5.8-fold and 4.4-fold higher than rFIX, respectively…” instead of “…transported 5.8-fold and 4.4-fold higher than rFIX-Tf, respectively…”.
Response: Thank you for your careful observation. Yes, the control is rFIX (BeneFix), not rFIX-Tf. When compared to FIX-Tf, FIX-Tf/SVSQ and FIX-Tf/Dithi were 2.4 and 1.6 folds higher than rFIX-Tf respectively. We have made the revision to avoid confusing.
Reviewer 2 Report
This study by Xie et al. describes a novel, orally available Transferrin receptor fused Factor IX to promote clotting in Hemophelia mice. The work is of significant interest to field and is well researched and described. While comprehensive, it would benefit from the following
1. How long does the effect of the administered Tf-Factor IX last in vivo? In other words, how long is the Factor IX bio-available post transfusion and/or oral delivery?
2. Can the authors use some other assays to confirm their findings, for instance intra-vital imaging to verify thrombus formation?
Author Response
Review #2 Comments and Suggestions for Authors
This study by Xie et al. describes a novel, orally available Transferrin receptor fused Factor IX to promote clotting in Hemophelia mice. The work is of significant interest to field and is well researched and described. While comprehensive, it would benefit from the following
How long does the effect of the administered Tf-Factor IX last in vivo? In other words, how long is the Factor IX bio-available post transfusion and/or oral delivery?Response: Thank you for the comment. We had conducted a preliminary study to test 5, 10, 15, and 20 min and found out that 18-20 min duration could give us the best results. Although we haven’t tested the exact duration of the drug effect, it can be reasonably estimated to be more than 20 min. In the future, a pharmacokinetic and pharmacodynamic study will be carried out to determine its accurate bioavailability and efficacy (including dose-response relationship and duration of action).
Can the authors use some other assays to confirm their findings, for instance intra-vital imaging to verify thrombus formation?
Response: Thank you for the suggestion. We agree it is important to validate the results by using other assays. Since this is a discovery stage proof-of-concept study, it is more essential to prove the feasibility of utilizing fusion protein to deliver protein drug orally. Therefore, we have not included other assays. For future drug development, we definitely need to verify our results using more assays such as the one you suggested to verify our results.